# The Effect of Natural Additives on the Composting Properties of Aliphatic Polyesters

**DOI:** 10.3390/polym12091856

**Published:** 2020-08-19

**Authors:** Malgorzata Latos-Brozio, Anna Masek

**Affiliations:** Institute of Polymer and Dye Technology, Lodz University of Technology, ul. Stefanowskiego 12/16, 90-924 Lodz, Poland; malgorzata.latos-brozio@dokt.p.lodz.pl

**Keywords:** biodegradable polyesters, plant substances, composting

## Abstract

Compounds of plant origin are used with polymers as functional additives. However, these substances often have biological (antimicrobial) activity. The bactericidal and fungicidal properties of natural additives can affect the composting process of biodegradable polymers. The scientific novelty of the manuscript is the investigation of the effect of the addition of herbal antimicrobial functional substances on the composting process of green polymers. The aim of the study is to analyze composting processes of biodegradable polymers polylactide (PLA) and polyhydroxyalkanoate (PHA) containing β-carotene, juglone, morin, and curcumin. As part of the research, six-month composting of materials was performed. At time intervals of one month, the weight loss of samples, surface energy, colour change, mechanical properties, and carbonyl indices (based on FTIR spectroscopy) of composted materials were examined. The research results showed that the addition of selected plant substances slightly slowed down the process of polymer composting. Slower degradation of samples with plant additives was confirmed by the results of mechanical strength tests and the analysis of changes in carbonyl index (CI). The CI analysis showed that PLA and PHA containing a natural additive degrade a month later than reference samples. However, PLA and PHA polyesters with β-carotene, juglone, morin, and curcumin were still very biodegradable.

## 1. Introduction

Biodegradable polymers such as polyhydroxyalkanoates (PHA) and polylactide (PLA) are becoming more popular due to their unique properties. Aliphatic polyesters PLA and PHA are easily composted and biodegradable, therefore they are an excellent alternative to commercial plastic packaging, especially for single use [1,2]. Biodegradation is the process of degradation caused by biological activity, particularly by enzymatic action, leading to a meaningful change in the chemical structure of the exposed polymeric material and resulting in the production of carbon dioxide, water, mineral salts (mineralization), and also new microbial cellular constituents (biomass). Biodegradable materials are plastics that undergo degradation due to the action of naturally occurring microorganisms, such as bacteria, fungi, and algae [3,4,5,6].

Nowadays, the development of green additives and fillers is increasing more and more to reduce the environmental impact of polymers. Natural compounds are increasingly used as functional substances in polymeric materials [7,8,9,10,11,12,13,14,15,16,17,18,19]. Phytocompounds, β-carotene, juglone, morin, and curcumin, were selected for this study.

B-carotene is a plant pigment from the carotenoids group. Carotenoids have been shown to act as antioxidants by quenching photosensitizers, interacting with singlet oxygen and scavenging peroxygen radicals. Moreover, these compounds exhibit a significant antimicrobial effect [20,21,22]. Another selected phytocompound, juglone, is a naphthoquinone type dye reported to be found in plants of the family *Juglandaceae* [23]. Juglone has been tested for phytotherapeutic applications due to its antiviral, antibacterial, and antifungal properties [23,24,25,26,27]. The next chosen natural compound was morin, belonging to the flavonoids group. Morin, like other flavonoids, has strong antioxidant properties and antibacterial and antifungal activity [28,29]. Curcumin was the last phytocompound introduced into polymers. It has been shown that this natural spice and common food colorant has, among other attributes, significant antifungal effect [30,31]. 

Natural substances (β-carotene, juglone, morin, and curcumin) have been used in the authors’ other manuscripts as stabilizers and indicators of aging time of polymers. However, in the cited publications, the influence of plant additives on the degradation processes of polymeric materials was not investigated [32,33,34,35,36,37]. Selected natural compounds, in addition to strong antioxidant and dyeing properties, also exhibit antimicrobial activity. The biological activity of compounds of plant origin may have a negative impact on the processes occurring during the biodegradation of polymers. Antimicrobial properties of phytocompounds can potentially slow down polymer composting and degradation processes.

Few literature references concern the analysis of the influence of functional plant substances on the degradation processes of eco-friendly polymers. In literature data, only Moraczewski et al. examined the effect of introducing natural anti-aging compounds of plant origin on polylactide composting [38]. Coffee, cocoa, and cinnamon extracts were added to polylactide in amounts of 0.5; 5, and 10 wt%. Samples were composted for 7, 14, 21, or 28 days. The authors showed that these plant extracts did not have negative effects on the industrial composting process. Coffee and cinnamon extracts even accelerated and intensified biodegradation of polylactide [38].

The aim of this study is to examine the effect of the addition of plant substances with antimicrobial activity on the composting process of biodegradable aliphatic polyesters. So far, the influence of β-carotene, juglone, morin, and curcumin on the composting of PLA and PHA has not been tested. The research presented in this manuscript develops scientific knowledge and helps to rectify the current dearth of information in the literature.

## 2. Materials and Methods

### 2.1. Reagents

The biodegradable polymeric materials used in this study were polylactide (PLA) and polymer P(3,4HB) 2001 from the polyhydroxyalkanoate group of polymers (PHA). Polylactide (PLA), Ingeo^TM^ Biopolymer 4043D PLA, was obtained from Nature Works^TM^ (Minnetonka, MN, USA) and had the properties: T_g_ = 55–60 °C, Tm = 145–160 °C, and melt flow index MFI = 6 g/10 min. Polyester (PHA) was produced by Simag Holdings LTD (Hong Kong, China) and had properties: P(3,4HB) containing 12 mol% 4-hydroxybutyrate, the average M_w_ was approximately 520 kDa, MVR = 15–20 g/10 min (assay conditions: temperature 170 °C, nominal load 2.16 kg), and a density of 1.25 g/cm^3^.

Phytochemicals: β-carotene ( ≥93%; Mw 536.87 g/mol; melting point range 178–179 °C), juglone (5-hydroxy-1,4-naphthoquinone, 97%; Mw 174.15 g/mol; melting point range 161–163 °C), morin (hydrate, ≥90%; Mw 302.24 g/mol; melting point range 299–300 °C), and curcumin ( ≥94%; Mw 368.38 g/mol; melting point 183 °C) were introduced into the polyester materials. All natural additives were purchased from Sigma-Aldrich (Darmstadt, Germany). Structural formulae of polymers and polyphenols are shown in Figure 1.

### 2.2. Method of Preparation of PLA and PHA with Phytochemicals

Dried (12 h, 50 °C) granulates of polymers PLA and PHA were mixed with 1 part by weight of β-carotene, juglone, morin, or curcumin and extruded using a laboratory extruder. Strip samples with a thickness of 1.6–1.8 mm were obtained. The temperature of the working chamber of the extruder was 180 °C for PLA (feed zone temperature 25 °C; cylinder zone temperature 190 °C; nozzle zone temperature 180 °C) and 160 °C for PHA (feed zone temperature 25 °C; cylinder zone temperature 165 °C; nozzle zone temperature 160 °C), screw rotation speed was 40 rpm, and extrusion pressure was 17 atm.

### 2.3. Method for Composting Polyester Samples

Polymer samples were placed in a ceramic vessel filled with garden soil with high peat content (soil pH 5.5–6.5) and left for 6 months. During the composting, a constant temperature of 30 °C and humidity (60%) were maintained and monitored. Samples were removed from the soil at intervals of 1 month for half a year. After removing soil residue with a brush, the samples were dried to constant weight and weighed, and then subjected to further analysis. The weight losses [%] were calculated based on the changes in the mass of the samples.

### 2.4. Measurement Methods

#### 2.4.1. Surface Free Energy

A goniometer OEC 15EC (DataPhysics Instruments GmbH, Filderstadt, Germany) was utilized to determine surface free energy of the polymers before and after composting. Surface free energy was calculated by the method of Owens, Wendt, Rabel, and Kaelble (OWRK) using software module SCA 20. Polar and disperse contributions to the surface energy and surface tension were combined by forming the sum of both parts, leading to (1) and (2):(1)$$\sigma_{l}=\sigma_{l}^{d}+\sigma_{l}^{p}$$
(2)$$\sigma_{S}=\sigma_{S}^{d}+\sigma_{S}^{p}$$
where $\sigma_{l}^{d}$ and $\sigma_{l}^{p}$ represent the disperse and polar parts of the liquid, while $\sigma_{S}^{d}$ and $\sigma_{S}^{p}$ stand for the respective contributions of the solid. 

Surface energy measurements were done based on the determination of contact angle. The measurements of contact angle were made for liquids with different polarities: distilled water, diiodomethane, and ethylene glycol. During the determination of surface energy on each of the three samples of one material, 10 contact angles were measured for each of the three liquids.

#### 2.4.2. Change of Color 

The change of color determinations were carried out using a CM-3600d spectrophotometer (Konica Minolta Sensing, Osaka, Japan). Color measurements were performed to measure the color change of the polymeric materials before and after composting. The result of the test is the color as described in the CIE-Lab space and the color in a system of three coordinates: L, a and b, where L is the lightness parameter (maximum value of 100, representing a perfectly reflecting diffuser, minimum value of zero representing the color black), a is the axis of red–green, and b is the axis of yellow–blue. The a and b axes have no specific numerical limits. The change of color, dE*ab, was computed according to Equation (3):(3)$$\mathrm{dE}*\mathrm{ab}\;=\;\sqrt{\left(\Delta a^{2})+(\Delta b^{2})+(\Delta L^{2}\right)}$$

Visual changes of color of samples before and after composting were recorded using a camera.

For change of color determination, 3 samples of each type of the polymeric material were prepared for each time interval of composting. Spectrophotometric determinations were done at 5 measuring points on each of the 3 control samples.

#### 2.4.3. Mechanical Properties

The mechanical properties tests presented in the manuscript were static tensile test. Mechanical properties of polymers were determined using a Zwick Roell Z005 test machine (Zwick Roell, Ulm, Germany) before and after composting. To determine mechanical properties, six control samples were cut out from extruded strips with a thickness of 1.6–1.8 mm and length of 150 mm. The measurement conditions were: a preload of 0.1 N and a test speed of 50 mm/min. The parameters: T_Fmax_, the maximum tensile stress [MPa], E_Fmax,_ the elongation at break for maximum tensile stress [%], σ, the tensile strength [MPa], and ε, the total elongation at break (just before the destruction of the sample) [%] were measured.

#### 2.4.4. Fourier Transform Infrared Spectroscopy (FTIR)

The measurements were carried out using a Thermo Scientific Nicolet 6700 FT-IR spectrometer (Thermo Fisher Scientific, Waltham, MA, USA) equipped with a diamond accessory, Smart Orbit ATR, for analyzing samples in the wave number range from 4000–400 cm^−1^. Samples were placed at the output of infrared beams. As the result of the study, oscillatory spectra were obtained, the analysis of which allows determination of the functional groups with which the radiation interacted.

For each of the analyzed samples, based on the FT-IR spectrum, a carbonyl index (CI) was calculated, according to Equation (4):(4)$$CI=\frac{I_{C=O}}{I_{C-H}}$$
where:

$I_{C=O}$—the intensity of the peak corresponding to the carbonyl groups C = O [-],

$I_{C-H}$—the intensity of the peak corresponding to the aliphatic carbon chains [-].

Carbonyl index (CI) is a measure of the number of carbonyl groups in the tested samples generated during composting.

## 3. Results and Discussion

Measuring the weight loss of a material after a certain composting time is the easiest and the most popular method for testing the approximate degree of biodegradation. In the process, some of the degraded material is converted into water, CO_2_ and other products, which results in a reduction of its mass. The biodegradation efficiency depends, not only on the conditions of composting, but also on the dimensions of the polymer sample.

Figure 2A,B shows the weight loss of PLA and PHA samples over 6 months of composting. After the first month of PLA composting, slight weight losses were found. The weight loss of the reference sample was 0.02% and of the samples containing natural additives from 0% (PLA/morin) to 0.08% (PLA/β-carotene). During the next four months of composting, the weight loss for PLA ranged from 0.14 to 0.38%. For samples containing plant substances, a weight loss from 0.10% to 0.84% was found. The greatest weight loss occurred after the sixth month of composing. After the last stage of composting, the weight loss of PLA was 2.84% and PLA with natural compounds from 0.60% (PLA/juglone) to 3.22% (PLA/β-carotene). For PHA samples, the weight loss after the first month of composting was 0.48% (reference sample) and from 0.04 to 0.36% for samples with natural additives. As a function of time, there was a gradual increase in weight loss of all samples, the greatest for the reference PHA (months 2–5: weight change from 0.54 to 3.91%) and for samples containing juglone (months 2–5: weight loss range from 0.56 to 3.35%). As in the case of PLA samples, the composition based on PHA also showed the greatest weight loss after 6 months of composting (PHA 5.64%, PHA with additives 0.40–3.67%). Higher mass losses of samples made using PHA polymer may result from the structure of the polymer matrix. As demonstrated by SEM studies published in another manuscript [39], the PHA samples were porous, while the PLA samples were compact and smooth. The high porosity of PHA materials can cause increased adsorption and absorption of water present during composting, as well as increased availability of material for microorganisms.

The behavior of PLA and PHA samples with curcumin and juglone seems to be interesting. PLA samples with juglone showed the lowest weight loss, while PHA samples with this plant additive showed weight losses similar to the reference PHA. Similarly, for the samples with curcumin–PLA/curcumin had the highest weight loss (greater than the reference PLA), and PHA/curcumin–low, similar to the samples with morin and β-carotene. This behavior of the samples may be related to the miscibility and solubility of specific natural additive in the polymer matrix of PLA or PHA. Moreover, the polarity of the PLA and PHA polymer matrices is different. This is evidenced by the surface energy values of polymers (PLA 42.05 mN/m^2^; PHA 37.63mN/m^2^). Due to the different polarity of the materials, the interactions of curcumin and juglon with polymers may be different. This may be the reason for a different effect of additives on the weight loss of PLA and PHA samples.

Reference samples of both polymers were more compostable than samples containing additives of plant origin. Due to fungicidal and bactericidal properties, polymer compositions with the addition of phytochemicals may show lower susceptibility to composting and biodegradation. Despite the lower compostability, the polyester materials were still degradable under composting conditions.

In the compost environment (elevated temperature, high humidity, presence of microorganisms), there are favorable conditions for the occurrence of two mechanisms of degradation of polymeric materials, i.e., hydrolytic degradation (a chemical process that causes chemical bonds in the polymer molecule to break) and enzymatic degradation (actions of microorganisms). In practice, the distribution of polymers under composting conditions is complex, which makes it impossible to identify only one mechanism responsible for this process. Degradation leads to a reduction in the molecular weight of all high molecular components of the polymer material, which in the case of one-component material, is associated with its distribution. In multi-component polymer materials, degradation of one of the components of a composite can only cause the loss of its cohesiveness and dispersion of other components in the environment, not complete degradation.

The second step in the study was to examine the color change of the samples after composting (Figure 3 and Figure 4). The change of color is the first sign of degradation of polymeric materials. Samples of both polyesters with the addition of substances of plant origin were characterized by a very pronounced color change, larger than the reference samples. When the color change factor dE*ab > 5, the colors are perceived as completely different. The largest color changes were found for samples containing B-carotene and juglone. Both B-carotene and juglone are substances of intense color and are successfully used as natural dyes, e.g., B-carotene is widely applied in the food industry. Juglone and substances from the quinone group of dyes are used, among other things, for dyeing fabrics. Composting samples can cause changes in the structure of the polymer matrix but can also initiate changes in the structure of substances of plant origin, e.g., cracking of C-C and C-H bonds, which will result in degradation of the plant additive and, as a consequence, a clear change in the color of the polymer material. Natural substances were added to various polymer matrices that differed from polarity, miscibility, affinity for specific additives. Therefore, the color change trends between PLA and PHA samples may be different. Visual changes of color of the samples during composting are shown in the photographs in Figure 3A and Figure 4A. Visual changes in the samples of both polyesters correspond to the spectrophotometric determination of the dE*ab coefficient.

An important parameter for polymeric materials is their surface energy, calculated on the basis of contact angle (Figure 5). The average contact angle for water for PLA and PLA with natural additives was 73°. A contact angle value below 90° indicates that the material has hydrophilic nature, and hence good compostability. Samples based on PHA polymer also had a hydrophilic surface character—the average contact angle for water was 72° (Table 1 and Table 2). The hydrophilic nature of both polyesters determines their good susceptibility to composting processes. For materials based on polylactide and polyhydroxyalkanoate, it was not found that the addition of substances of natural origin significantly affected the change in the surface energy of reference samples. Only the PLA/morin and PHA/juglone samples showed slightly lower surface energy compared to reference samples. The lower surface energy of the PLA/morin sample, compared to the standard PLA, may be due to the heterogeneity of the material. Morin dissolves poorly in polylactide. The heterogeneous structure of the sample may have reduced its surface energy compared to the reference sample. For materials made using PHA polymer, lower surface energy was found for the PHA/juglone sample relative to the PHA reference sample. The decrease in sample surface energy may be due to the juglone interaction with the PHA polymer. Similar behavior of juglone in PHA has been described elsewhere [33].

The reason for changes in the surface energy of samples during composting may be a change in the morphology of the materials. Composting, especially PHA, caused visible degradation of the samples’ surfaces (delicate cracks, roughness), which undoubtedly affected the changes in the surface energy values.

Plant substances, due to their fungicidal and bactericidal properties, can reduce the susceptibility of polymer compositions to composting and biodegradation. However, the analysis of the surface energy changes of the samples of both polyesters suggested that the surface energy values of the samples during composting were close to those for polyesters without plant substances, and indicated good degradability of the materials analyzed.

In the next stage of research, the effect of composting on the mechanical properties of selected polylactide and polyhydroxyalkanoate samples was analyzed. Table 3 summarizes the mechanical properties of PLA samples before and after 6 months of composting. In the first step of composting (after 1 month), polymer compositions PLA and PLA/curcumin showed an initial increase in the parameter T_Fmax_ [MPa] and σ [MPa] (maximum stress at break; tensile strength). Such results may indicate an increase in the crystalline phase content of samples after composting. After subsequent composting stages, PLA and PLA samples with curcumin showed a slow decrease in mechanical properties. For the PLA/juglone sample, the largest increase in Ts_max_ and σ was observed after 2 months of composting. In subsequent months, a decrease of the mechanical properties was noted. After 6 months of composting, it was found that the PLA reference had definitely lower values of all mechanical parameters than the samples with juglone and curcumin. For example, the σ value of the PLA sample was 36.2 MPa and for the PLA/juglone and PLA/curcumin samples were 55.2 MPa and 50.2 MPa, respectively. These results suggested that the addition of plant substances retards the composting of polylactide-based samples.

The degradation of biodegradable polymers, especially polylactide, is very specific. According to literature data [40], depending on the degradation conditions, there are different changes in the polylactide crystallinity. These changes also depend on the degree of crystallinity of the polymer. Changes in the polylactide crystallinity are part of the changes associated with the degradation of the polymer material, not a separate process. According to literature, when modelling chain cleavage induced crystallization in biodegradable PLLAs, it can be assumed that the crystal growth occurs much faster than the hydrolysis reaction. In semi-crystalline PLLAs, the amorphous polymer chains entrapped by the spherulites degrade much faster than the free amorphous polymer chains outside the spherules. The research results shown in publication [40] suggest that, during polylactide degradation, there is an increase in polymer crystallinity that accompanies the simultaneous decrease of its molar mass. Crystallization of polymers is, therefore, part of the degradation process, which is why there was an initial increase in mechanical properties during the composting study that may be associated with an increase in the degree of crystallinity of the samples.

Table 4 shows the mechanical properties of PHA-based samples before and after composting. Analysis of changes in the mechanical properties of PHA samples showed that the samples are clearly degraded after 3 months of composting. The significant decomposition of samples prevented the determination of the mechanical properties of all materials based on PHA. The samples were very brittle and cracked. As in the case of PLA samples, an initial increase in the degree of crystallinity of the PHA samples was observed (higher σ [MPa] tensile strength values), indicating the onset of material degradation. After subsequent composting time intervals, a clear decrease in the mechanical property parameters was observed. Polyester materials in which PHA was used as the polymer matrix were characterized by greater compostability than PLA samples.

During biodegradation, changes occur in the chemical structure of polymers that can be monitored by infrared spectroscopy (FT-IR). Degradation processes can cause detachment of substituents (giving signals in the IR spectrum) or cracking of C-C and C-H bonds in the main chain with the simultaneous formation of carbonyl, peroxide, and hydroxyl groups. If the polymer contains functional groups characterized by absorption in a specific range, by recording absorption spectra and analyzing changes in individual bands, it is possible to determine the degree of polymer degradation. The ratio of the intensity of a given band undergoing change to the bandwidth of a group not subject to change in the course of biodegradation is calculated. This ratio is called the group’s index (e.g., carbonyl index) and can be expressed as a percentage. Figure 6 summarizes the carbonyl indices of selected PLA and PHA samples calculated on the basis of FTIR spectra.

During composting, an increase in carbonyl indices of all samples was observed. For the PLA reference sample, an increase in the carbonyl index was found after 3 months of composting, while for the PLA/morin sample it was after 4 months of composting. The increase in carbonyl indices indicated a greater degree of degradation of the polymer compositions. The addition of morin to polylactide slowed down the composting process—a clear change in the carbonyl index was observed a month later than in the reference sample. 

The PHA polyester reference sample had higher carbonyl indices than the PLA sample, which indicated a greater susceptibility to degradation of the polyhydroxyalkanoate polymer. Significant structural changes, indicating the degradation of the PHA sample, were visible after the first month of composting, which translated into a more than 100% increase in the carbonyl index compared to the PHA/morin sample. Similar to the samples made using polylactide, for PHA samples also it was found that the addition of a plant-derived substance, morin, stabilized the polyester composition and retarded the composting process (slow increase of carbonyl index, pronounced after the fourth month of composting).

The appearance of OH hydroxyl groups in the FTIR spectra may indicate the hydrolytic degradation. However, in the spectra of the analyzed samples after composting, bands in range 3650–3450 cm^−1^ (corresponding to OH group) were not present. This may mean that enzymatic degradation was the dominant degradation mechanism of biodegradable materials.

## 4. Conclusions

Samples based on polyester from the group of polyhydroxyalkanoates, PHA, were more susceptible to composting, as evidenced by higher weight losses of samples and clear changes in carbonyl indexes calculated on the basis of FTIR spectra, as well as analysis of mechanical properties. Higher susceptibility of PHA samples to composting and biodegradation may be the result of the porous structure of the material, which ensured increased adsorption and absorption of water (hydrolytic degradation), as well as greater availability of material for microorganisms. Slower degradation was observed for PLA samples with a compact and smooth structure.

Polymer compositions with the addition of phytochemicals with antimicrobial properties were slightly less susceptible to composting and biodegradation, which was assessed by determining the weight loss of samples, carbonyl indices, and surface energy. Despite the lower susceptibility to composting, polyester materials with natural substances were still very degradable.

As a result of composting, not only the polymer matrix, but also plant additives underwent structural changes. Degradation of substances of natural origin caused, for example, the oxidation and cracking of their C-C and C-H bonds, which resulted in clear change of their color, as well as visual change throughout the polymeric materials. Composting can be an effective form of recycling polyester materials containing plant substances (β-carotene, juglone, morin, and curcumin) with biological activity.

## Figures and Tables

**Figure 1 polymers-12-01856-f001:**
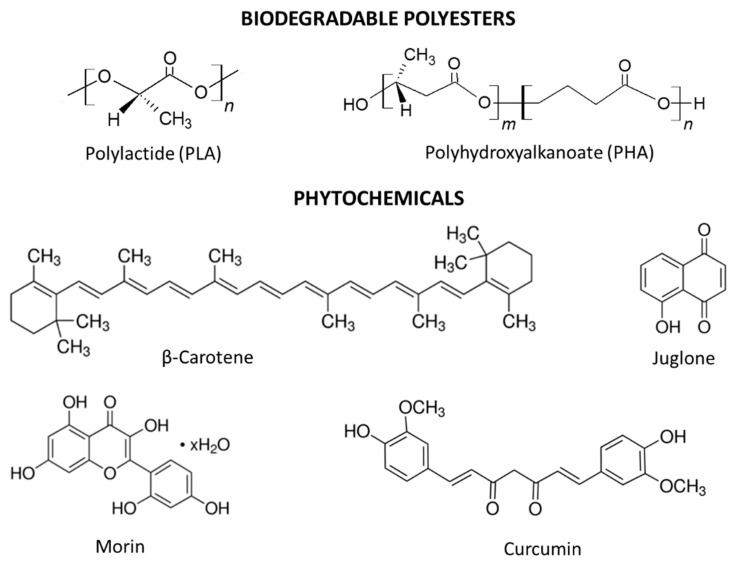
Structure of biodegradable polyesters and phytochemicals.

**Figure 2 polymers-12-01856-f002:**
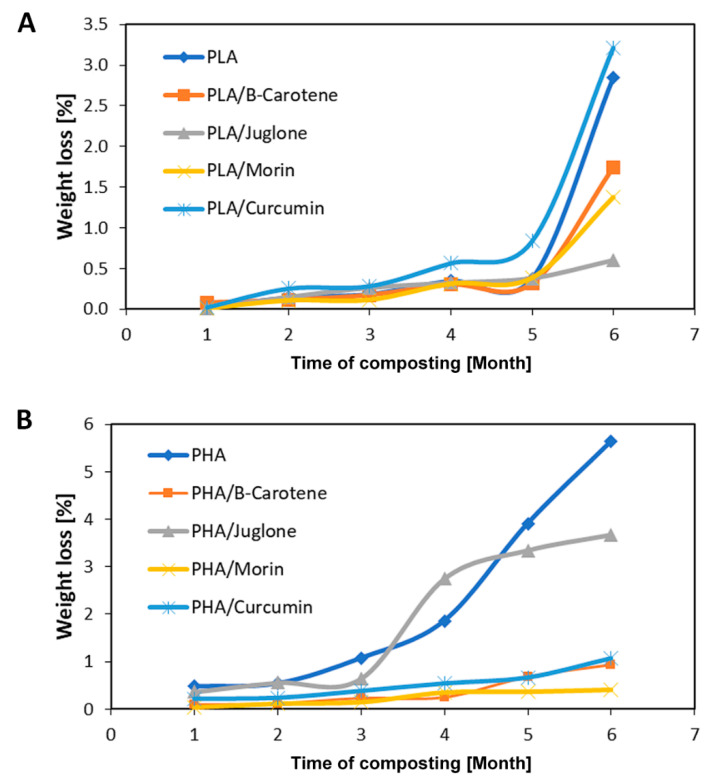
The weight loss of PLA (**A**) and PHA (**B**) samples over 6 months of composting.

**Figure 3 polymers-12-01856-f003:**
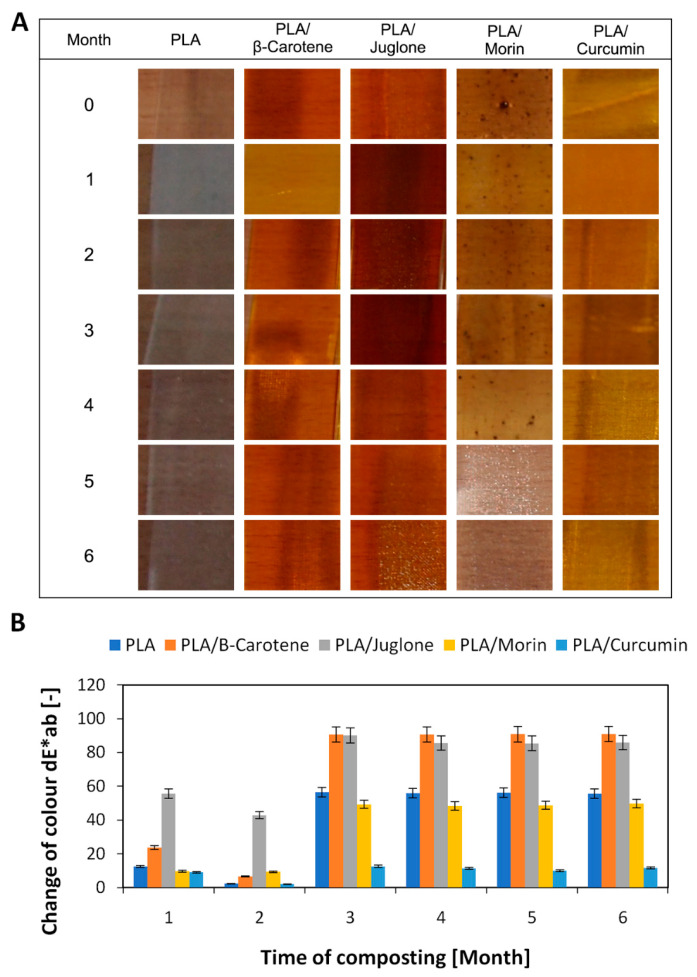
Visual changes of PLA samples (**A**) over 6 months of composting, confirmed by spectrophotometric determination of changes of color (**B**) in CIE-Lab space system.

**Figure 4 polymers-12-01856-f004:**
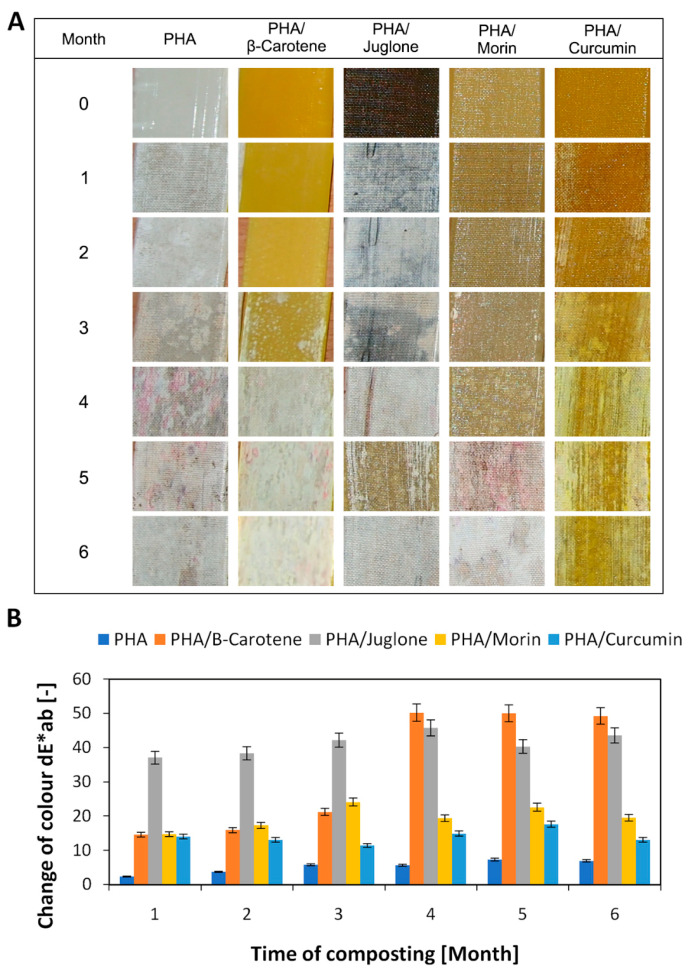
Visual changes of PHA samples (**A**) over 6 months of composting, confirmed by spectrophotometric determination of changes of color (**B**) in CIE-Lab space system.

**Figure 5 polymers-12-01856-f005:**
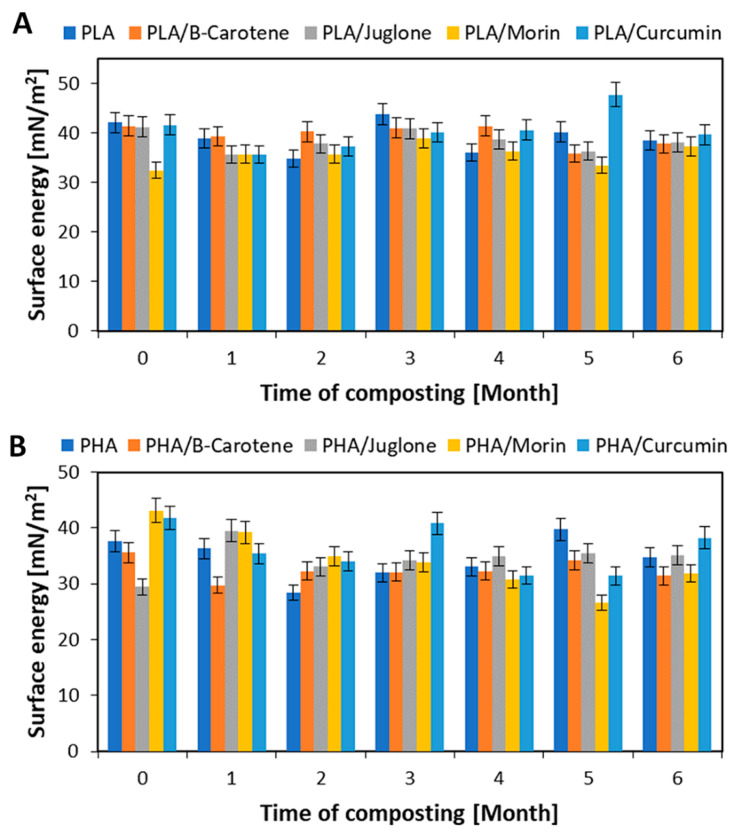
Changes in surface energy of PLA (**A**) and PHA (**B**) samples during composting.

**Figure 6 polymers-12-01856-f006:**
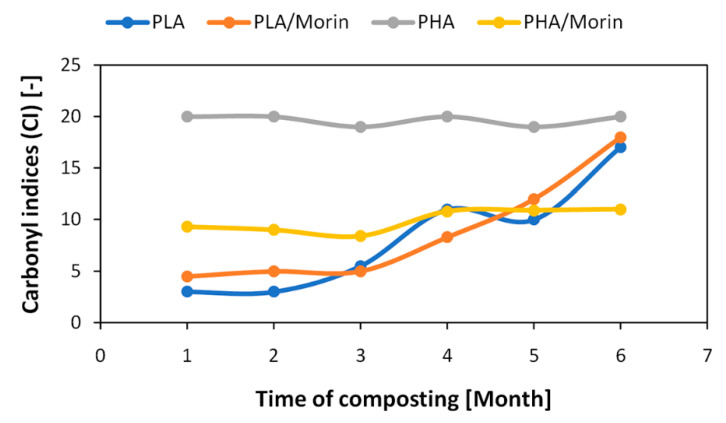
Carbonyl indices (CI) of selected PLA and PHA samples during composting.

**Table 1 polymers-12-01856-t001:** Contact angle obtained using water, diiodomethane, and ethylene glycol for PLA samples during 6 months of composting.

Liquid	Contact Angle during 6 Months of Composting [°]						
	0	1	2	3	4	5	6
**PLA**							
Water	61.3	77.1	70.8	76.8	75.6	75.2	66.8
Diiodomethane	36.7	36.8	53.7	27.6	47.3	38.7	42.2
Ethylene glycol	44.8	55.7	51.1	47.4	51.0	45.3	51.5
**PLA/B-Carotene**							
Water	69.9	80.5	73.4	76.7	77.9	81.2	78.5
Diiodomethane	35.3	39.8	38.2	38.9	32.9	45.4	38.2
Ethylene glycol	44.3	50.4	45.6	41.0	51.2	56.7	59.2
**PLA/Juglone**							
Water	74.5	73.7	68.7	68.9	76.8	85.2	77.9
Diiodomethane	37.2	48.0	46.9	39.9	39.8	45.1	40.2
Ethylene glycol	42.3	50.9	46.7	40.4	50.9	58.2	53.7
**PLA/Morin**							
Water	83.2	83.0	74.0	75.5	72.3	84.5	85.8
Diiodomethane	52.8	51.8	44.3	42.9	49.1	51.9	44.4
Ethylene glycol	58.8	44.1	57.7	43.6	48.2	56.4	55.8
**PLA/Curcumin**							
Water	77.6	84.4	70.0	75.1	78.8	74.2	76.7
Diiodomethane	32.0	51.2	46.3	34.9	34.6	38.1	35.1
Ethylene glycol	51.5	46.5	49.0	52.2	53.0	51.5	54.9

**Table 2 polymers-12-01856-t002:** Contact angle obtained using water, diiodomethane, and ethylene glycol for PHA samples during 6 months of composting.

Liquid	Contact Angle during 6 Months of Composting [°]						
	0	1	2	3	4	5	6
**PHA**							
Water	70.0	67.2	89.2	82.8	74.7	74.5	80.2
Diiodomethane	50.1	53.9	57.7	55.4	52.2	37.2	47.7
Ethylene glycol	41.4	48.1	70.6	55.8	58.0	49.9	51.1
**PHA/B-Carotene**							
Water	81.8	78.7	87.9	78.3	78.3	79.1	90.4
Diiodomethane	35.1	65.0	56.7	62.0	62.0	52.9	57.6
Ethylene glycol	76.6	53.5	53.4	45.2	45.2	49.3	50.6
**PHA/Juglone**							
Water	83.0	60.1	84.8	73.4	83.5	76.0	75.4
Diiodomethane	57.1	56.6	49.8	49.3	50.0	52.7	53.3
Ethylene glycol	64.9	44.3	62.8	57.4	52.5	44.3	49.8
**PHA/Morin**							
Water	70.7	83.1	71.8	77.9	80.1	87.6	88.4
Diiodomethane	32.2	40.3	43.8	51.4	60.5	60.8	52.3
Ethylene glycol	40.0	52.4	62.8	54.3	53.5	71.9	54.5
**PHA/Curcumin**							
Water	65.9	70.3	76.6	60.8	80.1	81.6	80.4
Diiodomethane	34.3	57.1	58.2	44.5	56.0	54.4	38.1
Ethylene glycol	45.5	44.3	42.1	44.0	57.1	60.3	45.2

**Table 3 polymers-12-01856-t003:** Mechanical properties of samples based on PLA over 6 months of composting (T_Fmax_ the maximum stress transferred by the sample [MPa], E_Fmax_ the elongation at break for maximum tensile stress [%], σ the tensile strength [MPa], and ε the total elongation at break [%]).

Sample	Time of Composting [Month]	T_Fmax_ [MPa]	E_Fmax_ [%]	σ [MPa]	ε [%]
**PLA**	0	44.9	6.4	39.6	8.1
	1	98.3	3.8	83.3	4.4
	2	40.5	3.6	37.6	3.9
	3	45.0	4.8	42.7	4.8
	4	38.1	4.0	37.9	4.0
	5	54.1	3.8	52.5	3.8
	6	37.0	2.9	36.2	2.9
**PLA/Juglone**	0	49.9	4.9	48.1	5.3
	1	46.7	3.7	56.3	3.8
	2	59.3	3.8	58.3	4.0
	3	52.1	4.8	51.3	4.9
	4	51.5	4.3	51.4	4.3
	5	52.9	4.8	51.2	4.8
	6	57.1	4.7	55.2	4.8
**PLA/Curcumin**	0	54.1	5.1	47.1	5.9
	1	97.2	4.3	19.3	18.1
	2	45.4	3.6	42.7	4.7
	3	53.0	4.8	51.2	4.9
	4	49.3	5.4	47.4	5.5
	5	58.5	4.9	57.9	5.3
	6	51.9	4.6	50.2	4.6

**Table 4 polymers-12-01856-t004:** Mechanical properties of PHA-based samples over 6 months of composting (T_Fmax_ the maximum stress transferred by the sample [MPa], E_Fmax_ the elongation at break for maximum tensile stress [%], σ the tensile strength [MPa], and ε the total elongation at break [%]).

Sample	Time of Composting [Month]	T_Fmax_ [MPa]	E_Fmax_ [%]	σ [MPa]	ε [%]
**PHA**	0	29.7	4.2	19.3	6.9
	1	21.1	2.3	21.1	2.3
	2	18.6	2.5	18.5	2.5
	3	10.9	2.3	10.5	2.3
	4	*	*	*	*
	5	*	*	*	*
	6	*	*	*	*
**PHA/Juglone**	0	32.5	3.8	6.4	5.8
	1	18.7	2.2	18.7	2.2
	2	11.9	1.6	11.9	1.6
	3	9.2	1.8	9.9	1.8
	4	*	*	*	*
	5	*	*	*	*
	6	*	*	*	*
**PHA/Curcumin**	0	30.0	4.4	5.99	6.1
	1	17.3	1.9	17.2	1.9
	2	4.09	1.4	3.87	1.5
	3	7.75	0.9	7.75	0.9
	4	*	*	*	*
	5	*	*	*	*
	6	*	*	*	*

* Significant degradation of the samples prevented the analysis of mechanical properties.

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
