# Peer review of "The Effect of Natural Additives on the Composting Properties of Aliphatic Polyesters"

_polymers, 2020, doi:10.3390/polym12091856_

Round 1
Reviewer 1 Report
The paper presents an experimental work on the effect of some natural additives derived from plants on the composting properties of two biodegradable polymers (PLA and PHA). The topic is attractive but the paper has some limitations regarding the discussion of the results and the Introduction. Here are my comments.
- Please change the title, which is not clear. A possible title could be “The effect of natural additives on the composting properties of aliphatic polyesters” .
- The abstract needs to be changed in the first part (rows 9-14) that should be shortened. Please, eliminate the sentence “Therefore, the submitted publication extends the literature data” .
- The introduction needs substantial changes in the organization and in the content. First of all, the problem is not well stated. In my opinion, the correct sequence in the introduction should : i) Biodegradable polymers and the issues related to their biodegradation; ii) PLA and PHA biodegradation (change the rows 49-59); iii) natural compounds used in general as additives for polymers and particularly for biodegradable polymers; iv) describe and highlight the differences respect to the previous authors works (references 12-17; 35), since the analysed additives are the same; v) aim of the work (improve the rows 30-31).
- Please complete the state of the art by citing recent works on the biodegradable polymer and additives as, for example: i) Materials 2019, 12(13), 2132; https://doi.org/10.3390/ma12132132; ii) Materials 2020, 13(9), 2200; https://doi.org/10.3390/ma13092200; iii) Materials 2019, 12, 2530; doi:10.3390/ma12162530; iv) Materials 2018, 11(1), 165; https://doi.org/10.3390/ma11010165
- Experimental: please add some details on the molecular weight and physical properties of the natural additives.
- Rows 93-94: please add all the operating temperatures of the extruder.
- Are the mechanical test carried out according to a standard? Why such a high speed of 50 mm/min?
- The results of weight loss should be deeply analysed. The authors should distinguish between the behavior in the first months and after six months of composting. The % reductions of weight loss should be reported in the analysis of the results. Why Curcumin has no effect on the PLA and a strong effect on PHA? Why Juglone has no effect on the PHA and a strong effect on PLA?
- A table with the contact angle obtained using the different liquids should be added in the text. As concerning the surface energy, it appears that after 6 months the values are within the experimental error, especially for PLA based samples.
- Table 1-2: express the mechanical results in the conventional way: Fmax, Greek letter epsilon for elongation at break, Greek letter sigma for tensile stress. Which is the difference between the EFmax the maximum elongation of the sample at break [%] and Eb the elongation at break [%]?
- Table 1-2: add the elastic modulus E (GPa) for the specimens.
- Based on the ontained results and particularly on FTIR results, it is possible to hypothesize to distinguish between the two mechanisms of hydrolytic degradation and enzymatic degradation?
- The authors should explain why it is important to increase the degradation time.
- Have the authors studied if some energy input (UV or ultrasound) can trigger or accelerate the biodegradation?
Author Response
Institute of Polymer and Dye Technology
Technical University of Lodz
90-924 Lodz, ul Stefanowskiego 12/16, Poland
Tel.: +48 42 631 32 23, Fax: +48 42 636 25 43
August 6, 2020
Polymers — Open Access Journal
Dear Professor,
We are resubmitting our revised paper entitled “The effect of composting aliphatic polyesters containing substances of plant origin” by Malgorzata Latos-Brozio and Anna Masek with a request to reconsider it for publication in "Polymers”.
We have carefully considered the Editor and Reviewers' comments. The manuscript was revised exactly according to these comments. The list of responses to the reviewer’s comments and corrections made in the manuscript is attached.
The manuscript has not been previously published, is not currently submitted for review to any other journal, and will not be submitted elsewhere before a decision is made by this journal.
For correspondence please use the following information:
corresponding author: Anna Masek
Institute of Polymer and Dye Technology
Technical University of Lodz
90-924 Lodz, ul Stefanowskiego 12/16, Poland
Tel.: +48 42 631 32 93
Fax: +48 42 636 25 43
e-mail: anna.masek@p.lodz.pl
Yours sincerely,
PhD, Dsc Anna Masek
Answers to reviewer #1 comments
Reviewer #1: The paper presents an experimental work on the effect of some natural additives derived from plants on the composting properties of two biodegradable polymers (PLA and PHA). The topic is attractive but the paper has some limitations regarding the discussion of the results and the Introduction. Here are my comments.:
Reviewer #1: Please change the title, which is not clear. A possible title could be “The effect of natural additives on the composting properties of aliphatic polyesters” .
Answer: We agree with the reviewer's comment. The title has been corrected.
Reviewer #1: The abstract needs to be changed in the first part (rows 9-14) that should be shortened. Please, eliminate the sentence “Therefore, the submitted publication extends the literature data .
Answer: The abstract has been corrected.
Reviewer #1: The introduction needs substantial changes in the organization and in the content. First of all, the problem is not well stated. In my opinion, the correct sequence in the introduction should: i) Biodegradable polymers and the issues related to their biodegradation; ii) PLA and PHA biodegradation (change the rows 49-59); iii) natural compounds used in general as additives for polymers and particularly for biodegradable polymers; iv) describe and highlight the differences respect to the previous authors works (references 12-17; 35), since the analysed additives are the same; v) aim of the work (improve the rows 30-31).
Answer: Thank you for your valuable comment. The introduction has been revised as suggested by the reviewer.
Reviewer #1: Please complete the state of the art by citing recent works on the biodegradable polymer and additives as, for example:
- i) Materials 2019, 12(13), 2132; https://doi.org/10.3390/ma12132132;
- ii) Materials 2020, 13(9), 2200; https://doi.org/10.3390/ma13092200;
iii) Materials 2019, 12, 2530; doi:10.3390/ma12162530;
- iv) Materials 2018, 11(1), 165; https://doi.org/10.3390/ma11010165
Answer: Literature references have been added.
Reviewer #1: Experimental: please add some details on the molecular weight and physical properties of the natural additives.
Answer: Details on the molecular weight and physical properties of the natural additives has been added.
Reviewer #1: Rows 93-94: please add all the operating temperatures of the extruder.
Answer: Extruder zone operating temperatures were added in manuscript.
Reviewer #1: Are the mechanical test carried out according to a standard? Why such a high speed of 50 mm/min?
Answer: The mechanical properties test was carried out in accordance with our internal standards. The test speed of 50 mm/min was selected experimentally as the most optimal for measuring analysed samples.
Reviewer #1: The results of weight loss should be deeply analysed. The authors should distinguish between the behavior in the first months and after six months of composting. The % reductions of weight loss should be reported in the analysis of the results. Why Curcumin has no effect on the PLA and a strong effect on PHA? Why Juglone has no effect on the PHA and a strong effect on PLA?
Answer: The description of the results of the weight loss of samples during composting has been extended in the manuscript.
After the first month of PLA composting, slight weight losses were found. The weight loss of the reference sample was 0.02% and of the samples containing natural additives from 0% (PLA/morin) to 0.08% (PLA/β-carotene). During the next four months of composting, the weight loss for PLA ranged from 0.14% to 0.38%. For samples containing plant substances, a weight loss from 0.10% to 0.84% was found. The greatest weight loss occurred after the sixth month of composing. After the last stage of composting, the weight loss of PLA was 2.84% and PLA with natural compounds from 0.60% (PLA/juglone) to 3.22% (PLA/β-carotene).
For PHA samples, the weight loss after the first month of composting was 0.48% (reference sample) and from 0.04% to 0.36% for samples with natural additives. As a function of time, there was a gradual increase in weight loss of all samples, the greatest for the reference PHA (months 2 - 5: weight change from 0.54% to 3.91%) and for samples containing juglone (months 2 - 5: weight loss range from 0.56% to 3.35%). As in the case of PLA samples, the composition based on PHA also showed the greatest weight loss after 6 months of composting (PHA 5.64%, PHA with additives 0.40% - 3.67%).
The behavior of PLA and PHA samples with curcumin and juglone seems to be interesting. PLA samples with juglone showed the lowest weight loss, while PHA samples with this plant additive showed weight losses similar to the reference PHA. Similarly, for the samples with curcumin - PLA/curcumin had the highest weight loss (greater than the reference PLA), and PHA/curcumin - low, similar to the samples with morin and β -carotene. This behavior of the samples may be related to the miscibility and solubility of specific natural additive in the polymer matrix of PLA or PHA. Moreover, the polarity of the PLA and PHA polymer matrices is different. This is evidenced by the surface energy values of polymers (PLA 42.05 mN/m2; PHA 37.63mN/m2). Due to the different polarity of the materials, the interactions of curcumin and juglon with polymers may be different. This may be the reason for a different effect of additives on the weight loss of PLA and PHA samples.
Reviewer #1: A table with the contact angle obtained using the different liquids should be added in the text. As concerning the surface energy, it appears that after 6 months the values are within the experimental error, especially for PLA based samples.
Answer: We agree with the reviewer's comment. Tables with the values of contact angles have been added in the manuscript.
Reviewer #1: Table 1-2: express the mechanical results in the conventional way: Fmax, Greek letter epsilon for elongation at break, Greek letter sigma for tensile stress. Which is the difference between the EFmax the maximum elongation of the sample at break [%] and Eb the elongation at break [%]?
Answer: The mechanical results were expressed as suggested by the reviewer. Parameters EFmax and Eb has been better defined in manuscript: EFmax, - elongation at break for maximum tensile stress; Eb/ε, - total elongation at break just before the destruction of the sample.
Reviewer #1: Table 1-2: add the elastic modulus E (GPa) for the specimens.
Answer: The mechanical properties tests presented in the manuscript were static tensile test. The testing machine used for mechanical analyzes and the available software allow to determine only the following parameters: TFmax, the maximum stress [MPa], EFmax, the elongation at break [%], TS, the tensile strength [MPa] and Eb, the elongation at break [%].
Unfortunately, we cannot determine the elastic modulus E (GPa) parameter.
Reviewer #1: Based on the ontained results and particularly on FTIR results, it is possible to hypothesize to distinguish between the two mechanisms of hydrolytic degradation and enzymatic degradation?
Answer: The appearance of OH hydroxyl groups in the FTIR spectra may indicate the hydrolytic degradation. However, in the spectra of the analyzed samples after composting, bands in range 3560-3450 cm-1 (corresponding to OH group) were not present. This may mean that enzymatic degradation was the dominant degradation mechanism of biodegradable materials.
Reviewer #1: The authors should explain why it is important to increase the degradation time.
Answer: The premise of the manuscript was to show the effect of the addition of bioactive natural substances on the composting of aliphatic polyesters. Due to the fact that phytochemicals have bactericidal and fungicidal properties, they can extend the composting time. However, the manuscript proved that the polymer compositions with plant compounds were still well degradable.
Reviewer #1: Have the authors studied if some energy input (UV or ultrasound) can trigger or accelerate the biodegradation?
Answer: Thank you for your valuable comment. In the studies presented in the manuscript, we did not test the effect of energy (UV or ultrasound) on the biodegradation of the samples. However, we think it is worth extending our research in the future.
We tested the impact of UV radiation on biodegradable materials in other our publications, such as:
- Masek, M. Latos-Brózio, The Effect of Substances of Plant Origin on the Thermal and Thermo-Oxidative Ageing of Aliphatic Polyesters (PLA, PHA), Polymers, 10(11) (2018) 1252, DOI:10.3390/polym10111252.
- Latos-Brózio, A. Masek, Effect of Impregnation of Biodegradable Polyesters with Polyphenols from Cistus Linnaeus and Juglans regia Linnaeus Walnut Green Husk, Polymers, 11 (2019), 669, DOI: 10.3390/polym11040669.
- Latos-Brózio, A. Masek, The application of (+)-catechin and polidatin as functional additives for biodegradable polyesters, International Journal of Molecular Sciences, 21 (2020) 414, DOI: 10.3390/ijms21020414.
- Latos-Brózio, A. Masek, Biodegradable Polyester Materials Containing Gallates, Polymers, 12 (2020) 677, DOI: 10.3390/polym12030677.
UV radiation accelerated the degradation of PLA and PHA-based materials.
Reviewer 2 Report
It is a good idea to use materials derived from nature as a material to control the rate of decomposition of biodegradable polymers.
However, some important discussions in the current paper need to be added.
1. In the weight loss experiment based on Figure 2, the author should explain the basic decomposition rate control mechanism. In particular, the properties of the four additives tend to be reversed in PLA and PHA, which needs to be explained.
2. What kind (status)of sample picture was taken in Figure 3-A and Figure 4-A?
3. Why is there a difference in color change between PLA and PHA? If the decomposition properties of additives are reflected, the trend should be the same.
4. What is the reason for the change of surface energy? For example, is there any influence of surface morphology?
5. The important thing to look at in tensile strength is not only the maximum strength, but also the tensile rate (elongation). The elastic modulus change of the material should be mentioned.
6. Is it possible to guarantee the stability of the material by blending in a high temperature environment?
Author Response
Institute of Polymer and Dye Technology
Technical University of Lodz
90-924 Lodz, ul Stefanowskiego 12/16, Poland
Tel.: +48 42 631 32 23, Fax: +48 42 636 25 43
August 6, 2020
Polymers — Open Access Journal
Dear Professor,
We are resubmitting our revised paper entitled “The effect of composting aliphatic polyesters containing substances of plant origin” by Malgorzata Latos-Brozio and Anna Masek with a request to reconsider it for publication in "Polymers”.
We have carefully considered the Editor and Reviewers' comments. The manuscript was revised exactly according to these comments. The list of responses to the reviewer’s comments and corrections made in the manuscript is attached.
The manuscript has not been previously published, is not currently submitted for review to any other journal, and will not be submitted elsewhere before a decision is made by this journal.
For correspondence please use the following information:
corresponding author: Anna Masek
Institute of Polymer and Dye Technology
Technical University of Lodz
90-924 Lodz, ul Stefanowskiego 12/16, Poland
Tel.: +48 42 631 32 93
Fax: +48 42 636 25 43
e-mail: anna.masek@p.lodz.pl
Yours sincerely,
PhD, Dsc Anna Masek
Answers to reviewer #2 comments
Reviewer #2: It is a good idea to use materials derived from nature as a material to control the rate of decomposition of biodegradable polymers. However, some important discussions in the current paper need to be added.
Reviewer #2: 1. In the weight loss experiment based on Figure 2, the author should explain the basic decomposition rate control mechanism. In particular, the properties of the four additives tend to be reversed in PLA and PHA, which needs to be explained.
Answer: The decomposition rate control mechanism was determined as the weight loss of the samples during composting. The weight losses [%] were calculated based on the changes in the mass of the samples. The detailed procedure for measuring the weight loss of samples is described in the methodology in section 2.3.
In addition, in the article we supplemented and explained the effect of plant additives on the weight loss of samples.
Reviewer #2: 2. What kind (status)of sample picture was taken in Figure 3-A and Figure 4-A?
Answer: Visual changes of colour of samples before and after composting were recorded using a camera.
Reviewer #2: 3. Why is there a difference in color change between PLA and PHA? If the decomposition properties of additives are reflected, the trend should be the same.
Answer: We agree with the reviewer. If the decomposition properties of additives are reflected, the trend should be the same. However, natural substances were added to various polymer matrices that differ, for example, in polarity, miscibility, affinity for specific additives. Therefore, the colour change trends between PLA and PHA samples may be different.
Reviewer #2: 4. What is the reason for the change of surface energy? For example, is there any influence of surface morphology?
Answer: Thank you for your important comment. The reason for changes in the surface energy of samples during composting may be a change in the morphology of the materials. Composting, especially PHA, caused visible degradation of the samples surfaces (delicate cracks, roughness), which undoubtedly affected the changes in the surface energy values.
Reviewer #2: 5. The important thing to look at in tensile strength is not only the maximum strength, but also the tensile rate (elongation). The elastic modulus change of the material should be mentioned.
Answer: The mechanical properties tests presented in the manuscript were static tensile test. The testing machine used for mechanical analyzes and the available software allow to determine only the following parameters: TFmax, the maximum stress [MPa], EFmax, the elongation at break [%], TS, the tensile strength [MPa] and Eb, the elongation at break [%].
Unfortunately, we cannot determine the elastic modulus E (GPa) parameter.
Reviewer #2: 6. Is it possible to guarantee the stability of the material by blending in a high temperature environment?
Answer: Higher environmental temperature may be a factor causing faster degradation of materials made of PLA and PHA polymers. We presented the influence of temperature on similar materials, among others in publications:
- Masek, M. Latos-Brózio, The Effect of Substances of Plant Origin on the Thermal and Thermo-Oxidative Ageing of Aliphatic Polyesters (PLA, PHA), Polymers, 10(11) (2018) 1252, DOI:10.3390/polym10111252.
- Latos-Brózio, A. Masek, The application of (+)-catechin and polidatin as functional additives for biodegradable polyesters, International Journal of Molecular Sciences, 21 (2020) 414, DOI: 10.3390/ijms21020414.
- Latos-Brózio, A. Masek, Biodegradable Polyester Materials Containing Gallates, Polymers, 12 (2020) 677, DOI: 10.3390/polym12030677.
Round 2
Reviewer 1 Report
Authors have addressed all my comments except for the determination of elatsic modulus.
I think that the paper can be suitable for the publication.